# The Effect of Gaseous Ozone Therapy in Conjunction with Periodontal Treatment on Glycated Hemoglobin Level in Subjects with Type 2 Diabetes Mellitus: An Unmasked Randomized Controlled Trial

**DOI:** 10.3390/ijerph17155467

**Published:** 2020-07-29

**Authors:** Biagio Rapone, Elisabetta Ferrara, Massimo Corsalini, Ilaria Converti, Felice Roberto Grassi, Luigi Santacroce, Skender Topi, Antonio Gnoni, Salvatore Scacco, Antonio Scarano, Maurizio Delvecchio

**Affiliations:** 1Department of Basic Medical Sciences, Neurosciences and Sense Organs, “Aldo Moro” University of Bari, 70121 Bari, Italy; feliceroberto.grassi@uniba.it (F.R.G.); gnoniantonio@gmail.com (A.G.); salvatore.scacco@uniba.it (S.S.); 2Complex Operative Unit of Odontostomatology, Hospital S.S. Annunziata, 66100 Chieti, Italy; igieneeprevenzione@gmail.com; 3Interdisciplinary Department of Medicine, University of Bari, 70121 Bari, Italy; massimo.corsalini@uniba.it; 4Department of Emergency and Organ Transplantation, Division of Plastic and Reconstructive Surgery, “Aldo Moro” University of Bari, 70121 Bari, Italy; ilaria.converti@gmail.com; 5Ionian Department (DJSGEM), “Aldo Moro” University of Bari, 70121 Bari, Italy; 6Department of Clinical Disciplines, School of Technical Medical Sciences, University A. Xhuvani, 3001 Elbasan, Albania; skender.topi@uniel.edu.al; 7Department of Oral Science, Nano and Biotechnology and CeSi-Met University of Chieti-Pescara, 66100 Chieti, Italy; ascarano@unich.it; 8Department of Metabolic and Genetic Diseases, Giovanni XXIII Children’s Hospital, 70126 Bari, Italy; mdelvecchio75@gmail.com

**Keywords:** type 2 Diabetes, periodontal disease, periodontitis, gaseous ozone therapy, periodontal treatment

## Abstract

Background: It is established that inflammation is involved in the pathogenesis of Type 2 Diabetes Mellitus (T2DM) by promoting insulin resistance and impaired beta cell function in the pancreas. Among the hypothesized independent risk factors implicated in the pathogenetic basis of disease, periodontal infection has been proposed to promote an amplification of the magnitude of the advanced glycation end product (AGE)-mediated upregulation of cytokine synthesis and secretion. These findings suggest an interrelationship between periodontal disease and type 2 diabetes, describing poor metabolic control in subjects with periodontitis as compared to nondiabetic subjects and more severe periodontitis in subjects with T2DM as compared to a healthy population, with a significant positive correlation between periodontal inflammatory parameters and glycated hemoglobin level. Results from clinical trials show that periodontal treatment is able to improve glycemic control in subjects with diabetes. Many therapeutic strategies have been developed to improve periodontal conditions in conjunction with conventional treatment, among which ozone (O_3_) is of specific concern. The principal aim of this trial was to compare the clinical effectiveness of an intensive periodontal intervention consisting of conventional periodontal treatment in conjunction with ozone gas therapy in reducing glycated hemoglobin level in type 2 diabetic patients and standard periodontal treatment. Methods: This study was a 12-month unmasked randomized trial and included 100 patients aged 40–74 years older, with type 2 diabetes mellitus diagnosed. All the patients received conventional periodontal treatment, or periodontal treatment in conjunction with ozone gas therapy in a randomly assigned order (1:1). The primary outcome was a clinical measure of glycated hemoglobin level at 3, 6, 9 and 12 months from randomization. Secondary outcomes were changes in periodontal inflammatory parameters. Results: At 12 months, the periodontal treatment in conjunction with ozone gas therapy did not show significant differences than standard therapy in decreasing glycated hemoglobin (HbA1C) level and the lack of significant differences in balance is evident. Conclusions: Although the change was not significant, periodontal treatment in conjunction with the gaseous ozone therapy tended to reduce the levels of glycated hemoglobin. The study shows a benefit with ozone therapy as compared to traditional periodontal treatment.

## 1. Introduction

Periodontal diseases encompass a wide variety of clinical phenotypes sharing the bacterial plaque as a common trigger, characterized by a progressive pathological change in the periodontium, which leads to the degeneration of supporting tissues of the teeth [1,2,3]. Based on the new classification of periodontal conditions, a wide range of periodontal diseases can be identified, characterized based on a multidimensional staging and grading system, indicating severity and extent of diseases, which include four categories that also involves systemic diseases and conditions that affect the periodontal supporting tissues, such as diabetes [1]. Under unhealthy general conditions, the peripheral inflammatory component may contribute to the exacerbation of a wide variety of disorders which may share a closely linked inflammatory etiology [4,5,6]. It is currently thought that periodontal infection plays a key role in mediating systemic inflammation in chronic diseases, such as type 2 diabetes [7,8,9,10], contributing significantly to the pathophysiology of disease [10]. The population with type II Diabetes (T2DM) presents a critical challenge in terms of periodontal condition. Diabetic patients are more prone to develop periodontal disease (estimated to be double the number than those in a healthy population), and are often affected by a severe form of periodontal infection, secondary to their underlying systemic alterations and are more likely to be resistant to periodontal treatment than individuals without diabetes [4,5]. Second, diabetic patients with periodontitis are known to be at elevated systemic inflammatory risk compared to the general population, making periodontal status control important for this group of patients [6,7]. The bidirectional relationship between periodontitis and diabetes has been widely discussed in literature [8,9,10,11], showing a significant improvement in both diseases after the eradication of periodontal infection. More direct evidence regarding the effects of periodontal disease on glycemic control of diabetic patients comes from intervention studies using periodontal therapy [12]. The chronic periodontal infection sustains a nidus for the inflammatory cascade, which results in the generalized release of inflammatory intermediates, playing a critical upstream role in blood glucose level increase [7]. The tagged structural relation and link defined between periodontal disease and diabetes is bidirectional [10]. Although periodontitis in patients with diabetes occurs as a secondary complication, it can occur in the setting of a variety of predisposing conditions to development of disease [11,12,13] On the other hand, diabetic condition appears to be capable of inducing the onset of periodontitis, sustaining the pathological process, establishing a vicious circle, and inducing a failed healing response in severe periodontal disease form [14]. It is well established that diabetic condition encourages the development of periodontal disease through a process involving the glucose-mediated advanced glycation end product (AGE) accumulation, that stimulates the gradual transformation of the subgingival microflora into a more pathogenic subgingival flora, responsible for the progression of the periodontal condition. On the other hand, periodontal infection has been proposed to promote an amplification of the magnitude of the advanced glycation end product (AGE)-mediated upregulation of cytokine synthesis and by chronic stimulus from LPS and products of periodontopathic organisms [15]. These metabolic end products are responsible of the degenerative changes usually observed in diabetic subjects [15,16].

Typically, periodontal infection exhibits a wide range of clinical severity that is often associated with severe diabetes mellitus [15,16,17,18]. The varying degrees of disorder are largely explained by heterogeneity in the causative components, and the disease in such patients may be modified by genetic or environmental factors [16]. In recent years, researchers have tried to confirm the benefits of periodontal treatment on blood glucose level in patients affected by type 2 diabetes [14,15,16,17]. It is likely that this advantage is mediated in part via effectively mitigating the exuberant inflammatory response. As such, a state of chronic inflammation associated with an elevation in inflammatory mediators may contribute to increase in glycated haemoglobin [12,16,19,20]. Many therapeutic strategies have been developed to improve periodontal conditions, among which ozone (O_3_) is of specific concern [21,22,23,24,25,26,27,28,29,30,31,32]. Gaseous ozone (O_3_-triatomic oxygen) derives by the photodissociation of molecular O2 into activated oxygen atoms and reaction with further oxygen molecules. Subsequently, the protonation of this transient radical generates HO3, which, in turn, decomposes to hydroxyl radical [21]. After further reactions, the conversion of O3 to the hydroxyl radical (OH-) occurs [22]. Ozone gas therapy has been proven to be a highly effective strategy for supporting periodontal treatment by the stimulation of nitric oxide (NO) production by endothelial cell nitric oxide synthase or other nitric oxide synthase isoforms, which exert local and systemic effects by inducing vasodilation [22], regulating leukocyte recruitment [23], scavenging endothelial-generated oxygen radicals [22,23], and preventing the upregulation of neutrophil CD11/CD18 [24,25,26]. For those reasons, we theorized that ozone therapy may be used as an adjuvant to conventional periodontal treatment in patients with type 2 diabetes. The principal aim of this trial was to investigate the hypothesis that the therapeutic effect of ozone in the eradication of periodontal infection might be more effective in the reduction in glycated hemoglobin (HbA1c) levels compared to conventional periodontal treatment.

## 2. Materials and Methods

### 2.1. Study Design

We conducted a single-center, unmasked randomized controlled study, using a design followed-up at 3, 6, 9 and 12 months from baseline, between February 2018 and January 2020. Eligible participants were identified using the routinely collected health records systems in primary care from electronic data files. In this 12-month randomized trial, 100 patients with type 2 diabetes were assigned in a 1:1 ratio to receive treatment with the ozone (closed-loop group) or a conventional therapy (control group).

### 2.2. Outcomes

The following clinical periodontal parameters were recorded at baseline, and at 3, 6, 9, and 12 months: plaque index (PI), probing pocket depth (PPD), bleeding on probing (BOP), and clinical attachment level (CAL). For the metabolic assessment, the blood samples, analyzed for glycated hemoglobin (HbA1c), were taken at baseline, and at the 3, 6, 9, and 12 months recall visits to monitor glucose control and periodontal status. HbA1C level was measured by the diabetes center. All data were collected from electronic searches from practice records.

### 2.3. Governance and Ethics

Ethical approval for this study was obtained from Ethics Committee Approval INTL_ALITMKCOOP/HealthMicroPath/HMM2019_IPM and was conducted according to Good Clinical Practice and the Declaration of Helsinki [27]. Written informed consent was obtained from all patients before the study.

### 2.4. Population

Potentially eligible participants were identified from patient records using automated searches, in terms of age and type 2 diabetes diagnosis. Subsequently, they were screened for periodontal disease and enrolled between June 2017 and October 2017. Inclusion criteria were age ≥18 years, type 2 diabetes diagnosis, periodontal disease, and those who had at least 25% of their teeth. The exclusion criteria were: a (I) concomitant disease such as cardiovascular disease, nephrologic conditions or other diabetic complications (II) and recent (less than 6 months) onset of heart failure; (III) known causes of heart failure; (IV) drug or alcohol abuse; (V) therapy with antibiotics within 6 months before the enrolment; (VI) contraindication to the treatment with ozone; (VII) pregnancy or lactation; (VIII) the presence of other conditions that might influence metabolic control; (IX) inability to understand the patient information or to give informed consent.

### 2.5. Sample Size

Sample size calculation was executed concerning the primary outcome variable, glycated hemoglobin (HbA1C). Power analysis calculations indicated a requirement of 96 patients, considering a clinical difference of 0.4% in the HbA1C between the two treatment methods with a 95% confidence interval (alpha = 0.05%).

### 2.6. Clinical Parameters

The following clinical parameters were recorded at six sites per tooth (mesio-buccal, mid-buccal, disto-buccal, mesio-palatal, mid-palatal, and disto-palatal), excluding the third molars, using a Williams periodontal probe (Nordent Manufacturing Inc., Elk Grove Village, IL, USA) calibrated in millimeters:Plaque Index (PI),Probing Pocket Depth (PDD),Bleeding on Probing (BOP)Clinical Attachment Level (CAL)

These indices represent a numerical value describing the periodontal status of the population. PI assumes a progression of gingivitis to pocket formation leading to advanced destruction. PDD and CAL are qualitative and quantitative criteria and a gingival and a periodontal component. BOP indicates inflammatory changes. All periodontal clinical examinations were collected at baseline and after 3, 6, 9 and 12 months by the same calibrated examiner.

### 2.7. Randomization

A web-based system was used to randomize patients 1:1 for conventional periodontal therapy/conventional periodontal therapy in conjunction with ozone therapy. In total, 100 periodontal patients with type 2 diabetes were randomly allocated to conventional periodontal treatment [28,29] or conventional periodontal treatment in conjunction with ozone therapy [28,29]. Neither patients nor investigators were masked to group allocation in this trial. The treatment assignment was unblinded for outcome.

### 2.8. Study Treatment

In two consecutive sessions, 100 patients were randomly assigned to receive treatment with conventional periodontal treatment and ozone gas therapy (Case group) or the standard periodontal treatment (Control group). Thus, all the patients received each treatment in a randomly assigned order. All subjects were instructed about oral hygiene regimen by a dental hygienist followed by non-surgical periodontal treatment, consisting of supragingival and subgingival removal of calculus and bacterial plaque (scaling and root planning, or SRP) and polishing, using ultrasonic and manual instruments. All treatments were performed under local anesthesia. Periodontal conditions were examined at baseline and recorded at each time point. After each treatment, the patients of Case group were treated with the gaseous ozone. The treatment was explained and demonstrated. The ozone delivery system, OZONE DTA (Ozone Generator, Sweden & Martina, Italy) was employed. The OZONE DTA is a device that takes in air and produces ozone gas. The ozone is then delivered via a hose into a disposable sterile cup at a concentration ranging from 10 to 100 μg/mL. Gaseous ozone was applied immediately after SRP at a fixed concentration of 2100 ppm with 80% oxygen four times for 30 s (every third day) for 1 week as per the manufacturer’s instructions. The full treatment consisted of three steps, as follows:Rinsing with ozonated water (1:3) for two minutes;Decontamination of the oral cavity;Drying the periodontal sites;Irrigating the periodontal pockets with 150 mL ozonized water;Applying gaseous ozone as previously described.

The participants were asked to return to the recruitment clinic every 3 months. At each appointment SRP was performed and all clinical parameters were recorded.

### 2.9. Statistics

Statistical analyses were conducted using SPSS software, (IBM SPSS Statistics 25, Armonk, NY, USA). Differences between means at baseline and post-treatment were evaluated using the Mann–Whitney U-test and the Wilcoxon signed-rank. Differences were considered significant at *p* ≤ 0.05. Quantitative measurements were expressed as mean + standard deviation (SD). Categorical data were presented as absolute frequencies and percent values. Difference between the two groups was determined by unpaired t-test for continuous variables and Fisher’s exact test for categorical data.

## 3. Results

A total of 100 patients underwent randomization; 50 were assigned to the ozone group, and 50 were assigned to the control group. Subjects were predominantly older. The mean age of the patients of control group was 55.05 ± 11.04. The mean age of the patients of case group was 57.55 ± 10.68. All 100 patients completed the trial. The glycated hemoglobin level ranged from 6.4 to 8.2%. The mean (±SD) percentage of the glucose level was within the target range decreased in the case group by 7% during the 12 months and decreased by 6% in the control group.

### Clinical Findings

Figure 1 shows the fluctuation in the HbA1C means over times. In the case group, treated with ozone therapy, the mean of HbA1C level decreased by 7% at 12 months after the therapy, as compared with control group in which the mean of HbA1C level decreased by 6% at 12 months after the conventional therapy. No significant statistical differences emerged. Figure 2 shows the value of PI means before and after the treatment. The mean PI in the case group, treated with ozone therapy, increased by 13% at 12 months after the therapy, as compared with control group in which the mean of PI remained stable at 12 months after the conventional therapy. The values of the BOP index are given in Figure 3. In the case group, the BOP level was significantly reduced by 92 % at 12 months after the therapy. In the control group, the mean BOP decreased by 92% at 12 months after the conventional therapy (Table 1). The levels of the PPD mean decreased significantly after treatment with ozone therapy (Figure 4), by 61% at 12 months after the therapy, as compared with control group in which the mean of PPD decreased by 41% at 12 months after the conventional therapy. The value of CAL is shown in Figure 5. In the case group, treated with ozone therapy, the mean CAL increased by 15% at 12 months after the therapy, as compared with control group in which the mean CAL was decreased by 17% at 12 months after the conventional therapy. Significant differences among the treatments were evident in CAL and Hba1C level at each time point (*p* > 0.05). At 3 and 6 months, the case group had slight reductions in glycated hemoglobin level compared with control group, however, there were no significant differences at 12 months (*p* > 0.05) (Table 2).

About the considered variables, the significance was observable for CAL at baseline and at 12 months, respectively: (z = −2.643; *p* = 0.007) and (z = −2.151; *p* = 0.030) at 12 months for PPD (z = −3.113; *p* = 0.001) and CAL (z = −2.705; *p* = 0.006) (Table 2).

## 4. Discussion

Metabolic abnormalities and long-term complications involving the eyes, kidneys, nervous system, vasculature, and periodontium characterize Diabetes Mellitus [29,30]. Furthermore, periodontitis has been recognized as an emergent “non-traditional” risk factor for Diabetes and has become the focus of extensive epidemiological research [3,4,5,6,7,8,9,10,11,12,13,14,15]. Systemic inflammatory processes, triggered by periodontal infection, induces the release of pro-inflammatory cytokines, which may directly impact the metabolic end product accumulation [7,8,9]. Several studies have investigated the effects of periodontal therapy on the fluctuation of glycated hemoglobin in diabetic patients, demonstrating an improvement in glycemic control, as determined by reductions in HbA1c values, after periodontal treatment. However, the study design, methodology and short-term results (3-6 months) of clinical trials provide conflicting data. Specifically, longitudinal observations are needed because of the chronic natural history of diabetes, involving organismic alterations that may influence the results of periodontal treatment. We, after periodontal treatment, assessed the short-term safety and efficacy of the gaseous ozone therapy with conventional non-surgical periodontal treatment for the control of glycated hemoglobin (HbA1c) in 100 periodontal patients with type 2 diabetes. Only a small number of studies have investigated the effectiveness of ozone therapy on periodontal parameters. The approach was expected to significantly improve the glycated hemoglobin (HbA1C) level [29,30,31,32,33,34,35,36,37,38]. Our results showed that both treatments resulted in significant improvements in all clinical and biochemical parameters, showing no significant difference between the conventional periodontal treatment and the treatment with a conjunction of gaseous ozone after 12 months. These results can be attributable to the localized effect of gaseous ozone. The periodontal parameter changes were similar among the groups. Our results are in accordance with a previous study conducted by Skurska et al. [39], which demonstrated a significant decrease in periodontal inflammatory indices after the association of periodontal treatment and gaseous ozone compared to the control group. In contrast, Uraz et al. [40] conducted a similar study showing that clinical parameters in patients with periodontitis did not improve with the adjunct of ozone therapy to conventional periodontal treatment. The authors observed that were no significant differences between two treatments groups for any of the microbiological, clinical and biochemical parameters.

## 5. Conclusions

To our knowledge, no studies have investigated the effect of gaseous ozone as support of periodontal treatment in diabetic patients. For this reason, it is difficult to measure the effectiveness of this therapy and to compare our results.

Although our findings suggested that gaseous ozone therapy is a noninvasive application that is safe and easy to perform, and may be a supporting treatment for periodontitis in patients with diabetes, having a beneficial effect on the glycated hemoglobin, it is unclear whether the results are generalizable to the larger population. Although there were statistically no differences between specific patient groups, there was a trend toward more effectiveness with gaseous ozone application following SRP. Therefore, currently available evidence suggests that gaseous ozone therapy is effective in the reduction in periodontal inflammatory status, but there are insufficient data to reach an evidence-based conclusion about the effectiveness of a reduction in glycated hemoglobin level in diabetic patients. In conclusion, the results of this study are suggestive, but one cannot arrive at a strong conclusion from the results of only one trial. Further longitudinal studies are needed to confirm the effectiveness of ozone application on glycated hemoglobin in diabetic patients.

## Figures and Tables

**Figure 1 ijerph-17-05467-f001:**
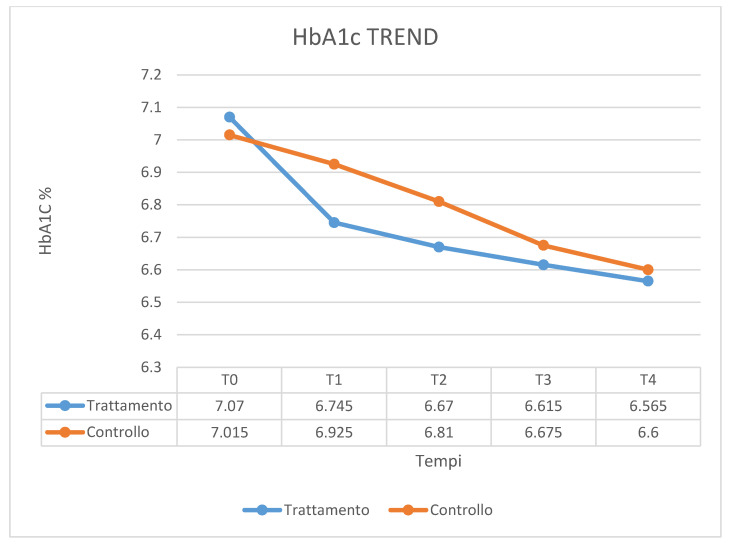
The fluctuation in the HbA1C means over time.

**Figure 2 ijerph-17-05467-f002:**
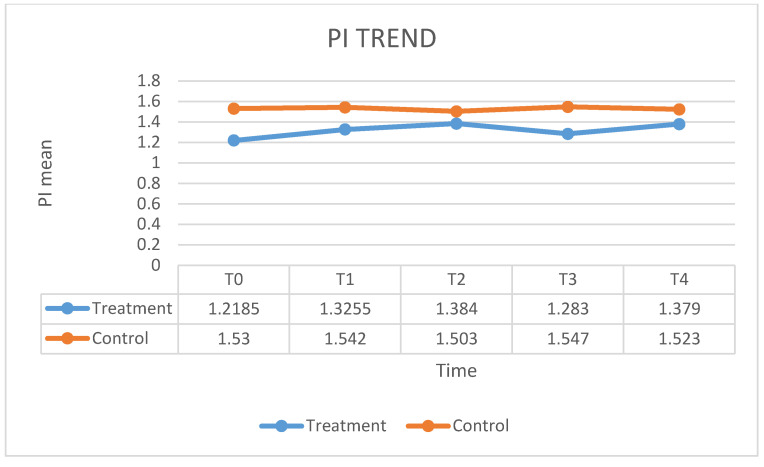
The fluctuation in the PI means over time.

**Figure 3 ijerph-17-05467-f003:**
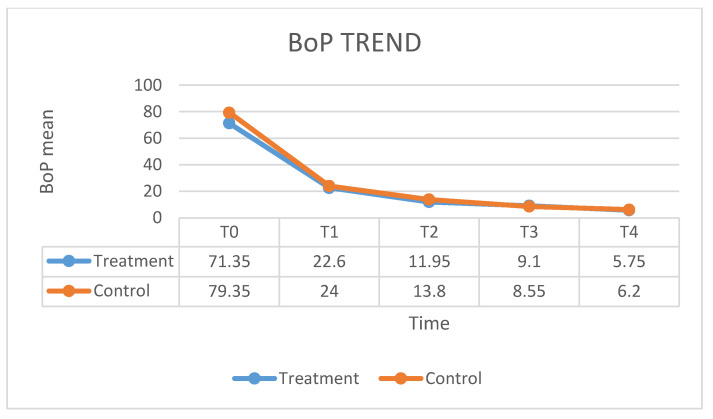
The fluctuation in the BOP means over time.

**Figure 4 ijerph-17-05467-f004:**
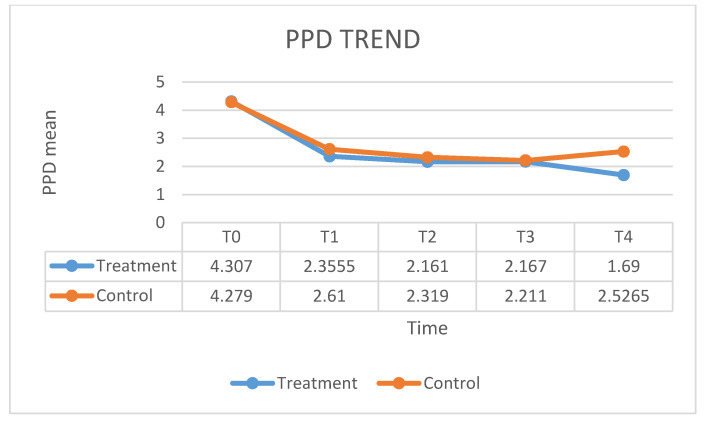
The fluctuation in the PPD means over time.

**Figure 5 ijerph-17-05467-f005:**
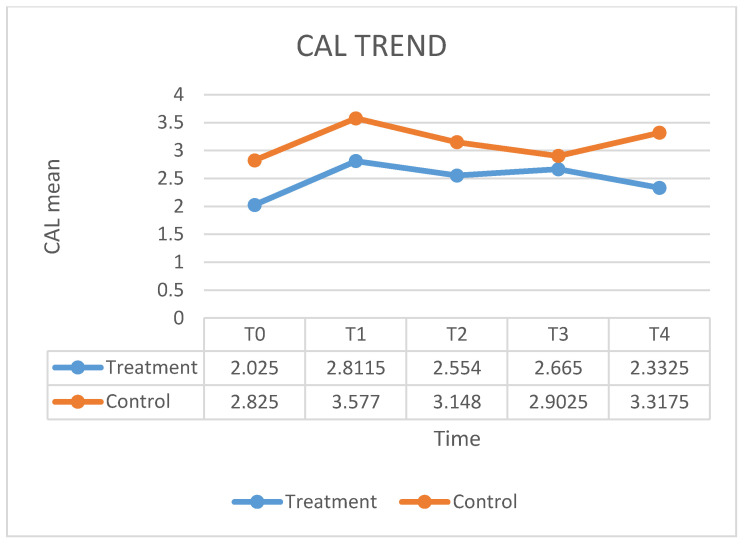
The fluctuation in the CAL means over time.

**Table 1 ijerph-17-05467-t001:** Clinical data about the control group.

	**PI T0**	**PI T1**	**PI T2**	**PI T3**	**PI T4**
Mean	77.35	22.2	14.05	9.2	6.8
SD	18.582	8.224	6.476	3.874	5.908
	**BoP T0**	**BoP T1**	**BoP T2**	**BoP T3**	**BoP T4**
Mean	79.35	24	13.8	8.55	6.2
SD	23.058	12.388	7.112	3.677	4.047
	**PPD T0**	**PPD T1**	**PPD T2**	**PPD T3**	**PPD T4**
Mean	4.279	2.61	2.319	2.211	2.526
SD	0.945	0.775	0.726	0.683	0.831
	**CAL T0**	**CAL T1**	**CAL T2**	**CAL T3**	**CAL T4**
Mean	2.825	3.577	3.148	2.902	3.317
SD	0.899	0.915	0.987	0.849	1.147
	**HbA1c T0**	**HbA1c T1**	**HbA1c T2**	**HbA1c T3**	**HbA1c T4**
Mean	7.015	6.925	6.81	6.675	6.6
SD	0.767	0.841	0.881	0.829	0.773

With respect to PI, PPD, BOP, CAL and HbA1C there were no registered significant differences (*p* > 0.05) at baseline between groups. Regarding the periodontal parameters, the values significantly decreased after the treatment in both the groups (*p* < 0.05).

**Table 2 ijerph-17-05467-t002:** Comparison of PI, PDD, CAL, and HbA1C values between SRP + OT and SRP-alone groups by Mann–Whitney *U* test.

Indices	U of Mann–Whitney	W of Wilcoxon	Z	Exact Sig. [2 × (1 − Taliled Sig.)]
PI T0	185,500	395,500	−0.397	0.698 ^b^
PI T1	166,000	376,000	−0.922	0.369 ^b^
PI T2	161,000	371,000	−1.060	0.301 ^b^
PI T3	190,500	400,500	−0.258	0.799 ^b^
PI T4	165,500	375,500	−0.949	0.355 ^b^
BoP T0	175,000	385,000	−0.691	0.512 ^b^
BoP T1	192,500	402,500	−0.203	0.841 ^b^
BoP T2	190,500	400,500	−0.258	0.799 ^b^
BoP T3	195,000	405,000	−0.136	0.904 ^b^
BoP T4	185,500	395,500	−0.394	0.698 ^b^
PPD T0	183,000	393,000	−0.460	0.659 ^b^
PPD T1	158,500	368,500	−1.124	0.265 ^b^
PPD T2	172,500	382,500	−0.744	0.461 ^b^
PPD T3	196,500	406,500	−0.095	0.925 ^b^
PPD T4	85,000	295,000	−3.113	0.001 ^b^
CAL T0	102,500	312,500	−2.643	0.007 ^b^
CAL T1	120,500	330,500	−2.151	0.030 ^b^
CAL T2	130,000	340,000	−1.894	0.060 ^b^
CAL T3	174,500	384,500	−0.690	0.495 ^b^
CAL T4	100,000	310,000	−2.705	0.006 ^b^
HbA1c T0	200,000	410,000	0.000	1.000 ^b^
HbA1c T1	178,000	388,000	−0.599	0.565 ^b^
HbA1c T2	172,500	382,500	−0.746	0.461 ^b^
HbA1c T3	188,000	398,000	−0.326	0.758 ^b^
HbA1c T4	199,000	409,000	−0.027	0.989 ^b^

^b^ indicates the positive ranks. *p* < 0.05, Mann–Whitney *U* test. PI: Plaque index, PPD: Periodontal Pocket Depth, BOP: Bleeding on probing, CAL: Clinical attachment Level, HbA1C: glycated hemoglobin.

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
