# Peer review of "The Effect of Gaseous Ozone Therapy in Conjunction with Periodontal Treatment on Glycated Hemoglobin Level in Subjects with Type 2 Diabetes Mellitus: An Unmasked Randomized Controlled Trial"

_ijerph, 2020, doi:10.3390/ijerph17155467_

Round 1
Reviewer 1 Report
General comments:
1) The clinical findings were inadequate to really assess the effects of ozone treatments, especially that already at time 0, CAL index was lower for ozone-treated patients as compared with that of control patient, weakening the significance of the differences.
2) The findings (Table 2) did not indicate significant differences in HbA1C levels in contrast to the claim made in the abstract.
3) The overall work appears quite preliminary as pointed by the authors in the conclusion “further longitudinal studies are needed to confirm the effectiveness of ozone application.”
Minor comments:
1) Abstract, lines 24-25, p1: Avoid the use of “believe” in “It is believed that inflammation is involved in the pathogenesis of Type 2 24 Diabetes Mellitus (T2DM) by promoting insulin resistance and impaired beta cell function in the 25 pancreas.”
2) Abstract, lines 26-28, p1: Rephrase since the sentence is a pleonasm in“periodontal infection has been proposed to play a role, promoting an exacerbation of inflammatory response.”
3) Abstract, lines 28-34, p1: Delete “On this view, the relationship between the two diseases may be understood in terms of interplay between distinct level of explanation.”
4) Abstract, lines 41-43, p1: Delete (PT) and (PT+OT) since thy were cited once in the abstract: “All the patients received conventional periodontal 41 treatment (PT), or periodontal treatment in conjunction to ozone gas therapy (PT + OT) in a 42 randomly assigned order (1:1).”
5) Abstract, line 45 p1 to line 48 p2: Replace HbA+C by its full name in “Results: At 12 months, the periodontal treatment in conjunction with ozone gas therapy was more effective than standard therapy in decreasing HbA1C level and 46 improving any secondary outcomes, but no significant difference between conventional periodontal 47 treatment and ozone therapy was recorded.”
6) Abstract, line 45 p1 to line 48 p2: The findings (Table 2) did not indicate significant differences in HbA1C levels in contrast to “Results: At 12 months, the periodontal treatment in conjunction with ozone gas therapy was more effective than standard therapy in decreasing HbA1C level and 46 improving any secondary outcomes, but no significant difference between conventional periodontal 47 treatment and ozone therapy was recorded.”
7) Introduction, line 70-71, p2: Add references in “Typically, periodontal infection exhibits a 70 wide range of clinical severity that often correlates with the level of glycated haemoglobin (HbA1c).”
8) Primary and Secondary Outcomes, lines 101-104: Add more information how glycated hemodlobin level was monitored in “The primary outcomes were the glycated hemoglobin level (HbA1C), as measured by venous blood monitoring, and changes in the values of periodontal parameters. Glycated hemoglobin level and periodontal clinical parameters level were measured and registered at baseline and each follow-1 up visit”
9) Primary and Secondary Outcomes, lines 105-106, p3: It was unclear what was the origin of secondary outcomes, specify PI, PDD, BOP and CAL and how they were determined in “Secondary outcomes included the differences in the PI, PDD, BOP, CAL, of both the groups. All data were collected from electronic searches from practice records.”
10) Governance & Ethics lines 107-110, p3 : Add more specific information to support “The trial was approved by the Ethics Committee of “Aldo Moro” University of Bari and was conducted according to Good Clinical Practice and the Declaration of Helsinki. Written informed consent was obtained from all patients.”
11) Clinical parameters, lines 128-136, p3: Add more information how PI, PDD, BOP, CAL were determined to support “The following clinical parameters were recorded at 6 sites per tooth (mesio-buccal, mid-buccal, disto-buccal, mesio-palatal, mid-palatal, and disto-palatal), excluding the third molars, using a Williams periodontal probe (Nordent Manufacturing Inc., IL, USA) calibrated in millimeters: Plaque Index (PI);Probing Pocket Depth (PDD), Bleeding on Probing (BOP), Clinical Attachment Level (CAL), All periodontal clinical examinations were collected at baseline and after 3, 6, 9 and 12 months 135 by the same calibrated examiner.”
12) Clinical parameters, lines 138-139, p4: Specify Web-based system or rephrase “Web-based system was used to randomize patients 1:1 for conventional periodontal 138 therapy/conventional periodontal therapy in conjunction with ozone therapy.”
13) Statistics, lines 168-169, p4: Add references to support “For the statistical analysis, IBM SPSS Statistics version 25 Windows package program was used. 168 The Mann-Whitney U-test and the Wilcoxon signed‐rank test were both used to compare the 169 continuous variables between the two groups on whether the statistical hypotheses were fulfilled.”
14) Results, line 176, p4: Figure1 was lacking in “Figure 1 shows the participant flow through the study”
15) Clinical findings, line 182, Table 1, p4: Specify T1, T2, T3 and T4 in Table 1
16) Clinical findings, line 185-8, Table 1, p5: Rephrase since CAL increased with the time in “As showed in Graph 1, Graph 2, Graph 3, Graph 4 and Graph 5 compared to that at baseline, all the parameters were significantly reduced at 3, 6, 9 and 12 months 187 after the therapy in both the groups (P < 0.05).”
17) Clinical findings, Delete Table 1 since Graph 1, Graph2, Graph3, Graph4 and Graph 5 since they are redundant with Table 1: Besides values of P1 in Table 1 are different than values of P1 in Graph2. Indicate bar errors in the figures. Translate into the English Trattamento and controllo in Graph1.
18) Clinical findings, lines 199-202, p7: Specify if the differences were significant to substantiate and check for the values “The Graph 1 shows the fluctuation in the HbA1C means over the times: in the case group, treated with ozone therapy, the mean of HbA1C level was decreased by 7% at 12 months after the therapy, as compared with control group in which the mean of HbA1C level was decreased by 6% at 12 months after the conventional therapy.”
19) Clinical findings, lines 202-205, p7: Graph 2 lacked bar errors and values of treated patients were already lower to those of control, making not sufficiently convincible to support “The Graph 2 shows the fluctuation in the PI means over the times: in the case group, treated with ozone therapy, the mean of PI was increased by 13% at 12 months after the therapy, as compared with control group in which the mean of PI was decreased by stable at 12 months after the conventional therapy.”
20) Clinical findings, 205-208, p7: Rephrase and shorten “The Graph 3 shows the fluctuation in the BOP means over the 205 times: in the case group, treated with ozone therapy, the mean of BOP was decreased by 92 % at 12 206 months after the therapy. In the control group, the mean of BOP was decreased by 92% at 12 months 207 after the conventional therapy. The Graph 4 shows the fluctuation in the PPD means over the times: 208 in the case group, treated with ozone therapy, the mean of PPD was decreased by 61% at 12 months. The Graph 5 shows the fluctuation in the CAL means 211 over the times: in the case group, treated with ozone therapy, the mean of CAL was increased by 15% 212 at 12 months after the therapy, as compared with control group in which the mean of CAL was 213 decreased by 17% at 12 months after the conventional therapy.”
21) Clinical findings, 214-217, p8: Values oh Hba1C levels were not significant according to Table 2 in contrast to the claims in “Significant differences among the treatments were evident in CAL and Hba1C level at each time point (P > 0.05). At 3 and 6 months, the case group had slight but significant reductions of glycated hemoglobin level compared with control group, however, there were no significant differences at 12 months (P > 0.05) (Table 2).” Besides mean CAL value of treated patients was already lower (2.025) as compared to that of control patients (2.825) at time 0 making the comparison inadequate.
22) Discussion, line 226, p8 to line 236, p9: Merge into the introduction “The population with type II Diabetes (T2DM) presents a critical challenge in terms of periodontal 226 condition. Diabetic patients are more prone to develop periodontal disease (is estimated to be double 227 the number in the healthy population), and often are affected by severe form of periodontal infection secondary to their underlying systemic alterations and are more likely to be resistant to periodontal treatment than individuals without diabetes [4, 5]. Second, diabetic patients with periodontitis are known to be at elevated systemic inflammatory risk compared to the general population, making periodontal status control an important for this group of patients [6, 7]. The bidirectional relationship between periodontitis and diabetes has been widely discussed in literature [22-26], showing a significant improvement of both diseases after the eradication of periodontal infection. More direct evidence regarding the effects of periodontal disease on glycemic control of diabetic patients comes from intervention studies using periodontal therapy [23].”
23) Discussion, lines 250-254, p9: Rephrase since it is contradictory. “Our results are according with a previous study conducted by Skurska et al. [31], which demonstrated a significant decrease of periodontal inflammatory indices after the association of periodontal treatment and gaseous ozone, but did not register a greater improvement in these parameters in patients who underwent only scaling and root ”
24) Discussion, lines 254-260, p9: Rephrase since two sentences are contradictory in “Interestingly, Uraz et al [32]. conducted a similar study showing that clinical parameters in patients with periodontitis did not improve with the adjunct of ozone therapy to conventional periodontal treatment. The authors observed that were no significant differences between two treatments groups for any of the microbiological, clinical and biochemical parameters. To our knowledge, no studies have investigated the effect of gaseous ozone as support of periodontal treatment in diabetic patients. For this reason, it is difficult to measure the effectiveness of this therapy and to compare our results.”
25) Conclusion, lines 262-266: The findings were insufficiently convincible to support “In conclusion, gaseous ozone therapy is a noninvasive application, safe and easy to perform, and may be a supporting treatment for periodontitis, having a beneficial effect on the periodontium. Gaseous ozone application following SRP may contribute to improvements in the glycated hemoglobin level, but further longitudinal studies are needed to confirm the effectiveness of ozone application.”
Author Response
Reviewer #1
1)The clinical findings were inadequate to really assess the effects of ozone treatments, especially that already at time 0, CAL index was lower for ozone-treated patients as compared with that of control patient, weakening the significance of the differences.
-The Clinical attachment level (CAL) index is only one of the periodontal indices, depending on several factors, including the gingival phenotype, the age and gender, etc. Further, it is possible to have a low value of CAL and a concomitant higher level of periodontal pocket depth index. It is determinant the fluctuation of this value and its variability over the time. The characteristics of periodontal disease represent an heterogeneous phenomena, and the clinical manifestations of the disease caused by interactions between the host and agents may vary according to the site and tooth and may even vary within the same individual and longitudinally over time.
2) The findings (Table 2) did not indicate significant differences in HbA1C levels in contrast to the claim made in the abstract.
-This was an error. It has been corrected. Indeed, although in contrast with to the claim in the abstract, it was indicated also “...but no significant difference between conventional periodontal treatment and ozone therapy was recorded”. Our concern was to explain the interindividual differences.
3) The overall work appears quite preliminary as pointed by the authors in the conclusion “further longitudinal studies are needed to confirm the effectiveness of ozone application.”
-The phrase it has been corrected, to explain the reason of our ascertainment. The ozone therapy is sure effective
Minor comments:
- Abstract, lines 24-25, p1: Avoid the use of “believe” in “It is believed that inflammation is involved in the pathogenesis of Type 2 24Diabetes Mellitus (T2DM) by promoting insulin resistance and impaired beta cell function in the 25 pancreas.”
-It has been corrected with: established.
2) Abstract, lines 26-28, p1: Rephrase since the sentence is a pleonasm in “periodontal infection has been proposed to play a role, promoting an exacerbation of inflammatory response.”
-It has been rephrased as follows:
Among the hypothesized independent risk factors implicated in the pathogenetic basis of disease, periodontal infection has been proposed to promote an amplification of the magnitude of the advanced glycation end product (AGE)-mediatedupregulation of cytokine synthesis and secretion.
3) Abstract, lines 28-34, p1: Delete “On this view, the relationship between the two diseases may be understood in terms of interplay between distinct level of explanation.”
- It has been deleted
4) Abstract, lines 41-43, p1: Delete (PT) and (PT+OT) since thy were cited once in the abstract: “All the patients received conventional periodontal 41 treatment (PT), or periodontal treatment in conjunction to ozone gas therapy (PT + OT) in a 42 randomly assigned order (1:1).”
-It has been deleted
5) Abstract, line 45 p1 to line 48 p2: Replace HbA+C by its full name in “Results: At 12 months, the periodontal treatment in conjunction with ozone gas therapy was more effective than standard therapy in decreasing HbA1C level and 46 improving any secondary outcomes, but no significant difference between conventional periodontal 47 treatment and ozone therapy was recorded.”
-It has been replaced with its full name.
6) Abstract, line 45 p1 to line 48 p2: The findings (Table 2) did not indicate significant differences in HbA1C levels in contrast to “Results: At 12 months, the periodontal treatment in conjunction with ozone gas therapy was more effective than standard therapy in decreasing HbA1C level and 46 improving any secondary outcomes, but no significant difference between conventional periodontal 47 treatment and ozone therapy was recorded.”
-This was an error. It has been corrected. Indeed, although in contrast with to the claim in the abstract, it was indicated also “...but no significant difference between conventional periodontal treatment and ozone therapy was recorded”. Our concern was to explain the interindividual differences.
7) Introduction, line 70-71, p2: Add references in “Typically, periodontal infection exhibits a 70 wide range of clinical severity that often correlates with the level of glycated haemoglobin (HbA1c).”
-It has been added as follows:
Xue Jiang, Yanlin Zhu, Zhaoying Liu, Zilu Tian, Song Zhu, Association between diabetes and dental implant complications: a systematic review and meta-analysis, Acta Odontologica Scandinavica, 10.1080/00016357.2020.1761031, (1-10), (2020).
Mayra Moura FRANCO, Mariana Mader Miranda MORAES, Poliana Mendes DUARTE, Marcelo Henrique NAPIMOGA, Bruno Braga BENATTI, Glycemic control and the production of cytokines in diabetic patients with chronic periodontal disease, RGO - Revista Gaúcha de Odontologia, 10.1590/1981-863720170001000063063, 65, 1, (37-43), (2017).
- Li X, Kolltveit KM, Tronstad L, Olsen I. Systemic diseases caused by oral infection. Clin Microbiol Rev. 2000;13(4):547-558. doi:10.1128/cmr.13.4.547-558.2000
- Aldridge J P, Lester V, Watts T L, Collins A, Viberti G, Wilson R F. Single-blind studies of the effects of improved periodontal health on metabolic control in type 1 diabetes mellitus. J Clin Periodontol. 1995;22:271–275.
- Christgau M, Palitzsch K D, Schmalz G, Kreiner U, Frenzel S. Healing response to non-surgical periodontal therapy in patients with diabetes mellitus: clinical, microbiological, and immunologic results. J Clin Periodontol. 1998;25:112–124.
8) Primary and Secondary Outcomes, lines 101-104: Add more information how glycated hemodlobin level was monitored in “The primary outcomes were the glycated hemoglobin level (HbA1C), as measured by venous blood monitoring, and changes in the values of periodontal parameters. Glycated hemoglobin level and periodontal clinical parameters level were measured and registered at baseline and each follow-1 up visit”
-It has been specified that HbA1C level has been measured by the diabetes center.
“The following clinical periodontal parameters were recorded at baseline, and at the 3, 6, 9, and 12 months: plaque index (PI), probing pocket depth (PPD), bleeding on probing (BOP), and clinical attachment level (CAL). For the metabolic assessment, the blood samples were analyzed for glycated hemoglobin (HbA1c) was taken at baseline, and at the 3, 6, 9, and 12 months recall visit to monitor glucose control and periodontal status. HbA1C level was been measured by the diabetes center. All data were collected from electronic searches from practice records.”
9) Primary and Secondary Outcomes, lines 105-106, p3: It was unclear what was the origin of secondary outcomes, specify PI, PDD, BOP and CAL and how they were determined in “Secondary outcomes included the differences in the PI, PDD, BOP, CAL, of both the groups. All data were collected from electronic searches from practice records.”
- The origin and determination of these indices are specified in the next section, named “Clinical parameters”
10) Governance & Ethics lines 107-110, p3: Add more specific information to support “The trial was approved by the Ethics Committee and was conducted according to Good Clinical Practice and the Declaration of Helsinki. Written informed consent was obtained from all patients.”
-It has been added as follows:
Ethical approval for this study was obtained from Ethics Committee Approval INTL_ALITMKCOOP/HealthMicroPath/HMM2019_IPM and was conducted according to Good Clinical Practice and the Declaration of Helsinki [27]. Written informed consent was obtained from all patients before the study.
11) Clinical parameters, lines 128-136, p3: Add more information how PI, PDD, BOP, CAL were determined to support “The following clinical parameters were recorded at 6 sites per tooth (mesio-buccal, mid-buccal, disto-buccal, mesio-palatal, mid-palatal, and disto-palatal), excluding the third molars, using a Williams periodontal probe (Nordent Manufacturing Inc., IL, USA) calibrated in millimeters: Plaque Index (PI);Probing Pocket Depth (PDD), Bleeding on Probing (BOP), Clinical Attachment Level (CAL), All periodontal clinical examinations were collected at baseline and after 3, 6, 9 and 12 months 135 by the same calibrated examiner.”
-It has been added as follows:
These indices represent a numerical value describing the periodontal status of the population. These indices represent a numerical value describing the periodontal status of the population. PI assumes a progression of gingivitis to pocket formation leading to advanced destruction. PDD and CAL are qualitative and quantitative criteria and a gingival and a periodontal component. BOP indicates inflammatory changes.
12) Clinical parameters, lines 138-139, p4: Specify Web-based system or rephrase “Web-based system was used to randomize patients 1:1 for conventional periodontal 138 therapy/conventional periodontal therapy in conjunction with ozone therapy.”
-It has been rephrased as follows:
Statistical analyses were conducted using SPSS software, (IBM SPSS Statistics 25). Differences between means at baseline and post-treatment were evaluated using the The Mann-Whitney U-test and the Wilcoxon signed‐rank. Differences were considered significant at P ≤ 0.05. Quantitative measurements were expressed as mean + standard deviation (SD). Categorical data were presented as absolute frequencies and percent values. Difference between the two groups was determined by unpaired t-test for continuous variables and Fisher's exact test for categorical data.
13) Statistics, lines 168-169, p4: Add references to support “For the statistical analysis, IBM SPSS Statistics version 25 Windows package program was used. 168 The Mann-Whitney U-test and the Wilcoxon signed‐rank test were both used to compare the 169 continuous variables between the two groups on whether the statistical hypotheses were fulfilled.”
-It has been added.
[Akshar Bharathi et al.,1998] Akshar Bharathi, Rajeev Sangal and Sushma M Bendre: Some Observations Regarding Corpora of Indian Languages. Proc. of Int. Conf. on Knowledge-Based Computer Systems (KBCS-98), 17-19 Dec 1998, NCST, Mumbai
[Daniel jurafsky and James H.Martin] Daniel jurafsky & James H.Martin : An introduction to NLP ,computational linguistics, and speech recognition,(page no –223)
[G S Lehal et al.,1998] G S Lehal, Renu Dhir, Ritu Lehal: Corpus based statistical analysis of printed Punjabi text. Proc. of Int. Conf. on Knowledge-Based Computer Systems (KBCS-98), 17-19 Dec 1998, NCST, Mumbai
14) Results, line 176, p4: Figure1 was lacking in “Figure 1 shows the participant flow through the study”
-It was an error. It has been deleted.
15) Clinical findings, line 182, Table 1, p4: Specify T1, T2, T3 and T4 in Table 1
- It has been specified.
16) Clinical findings, line 185-8, Table 1, p5: Rephrase since CAL increased with the time in “As showed in Graph 1, Graph 2, Graph 3, Graph 4 and Graph 5 compared to that at baseline, all the parameters were significantly reduced at 3, 6, 9 and 12 months 187 after the therapy in both the groups (P < 0.05).”
-It has been rephrased as follows:” Regarding the periodontal parameters, the values were significantly decreased after the treatment in both the groups (P < 0.05).”
17) Clinical findings, Delete Table 1 since Graph 1, Graph2, Graph3, Graph4 and Graph 5 since they are redundant with Table 1: Besides values of P1 in Table 1 are different than values of P1 in Graph2. Indicate bar errors in the figures. Translate into the English Trattamento and controllo in Graph1.
-It has been deleted Table 1 and it has been translated into the English Trattamento and controllo in Graph1.
18) Clinical findings, lines 199-202, p7: Specify if the differences were significant to substantiate and check for the values “The Graph 1 shows the fluctuation in the HbA1C means over the times: in the case group, treated with ozone therapy, the mean of HbA1C level was decreased by 7% at 12 months after the therapy, as compared with control group in which the mean of HbA1C level was decreased by 6% at 12 months after the conventional therapy.”
-It has been added that “No significant statistical differences emerged.”
19) Clinical findings, lines 202-205, p7: Graph 2 lacked bar errors and values of treated patients were already lower to those of control, making not sufficiently convincible to support “The Graph 2 shows the fluctuation in the PI means over the times: in the case group, treated with ozone therapy, the mean of PI was increased by 13% at 12 months after the therapy, as compared with control group in which the mean of PI was decreased by stable at 12 months after the conventional therapy.”
- Our aim was to provide a framework for evaluating also small area variations about periodontal indices, regardless of the severity of periodontal condition.
20) Clinical findings, 205-208, p7: Rephrase and shorten “The Graph 3 shows the fluctuation in the BOP means over the 205 times: in the case group, treated with ozone therapy, the mean of BOP was decreased by 92 % at 12 206 months after the therapy. In the control group, the mean of BOP was decreased by 92% at 12 months 207 after the conventional therapy. The Graph 4 shows the fluctuation in the PPD means over the times: 208 in the case group, treated with ozone therapy, the mean of PPD was decreased by 61% at 12 months. The Graph 5 shows the fluctuation in the CAL means 211 over the times: in the case group, treated with ozone therapy, the mean of CAL was increased by 15% 212 at 12 months after the therapy, as compared with control group in which the mean of CAL was 213 decreased by 17% at 12 months after the conventional therapy.”
-It has been corrected as follows:
The values of BOP index is given in Graph 3. In the case group, the BOP level was significantly reduceed by 92 % at 12 months after the therapy. In the control group, the mean of BOP was decreased by 92% at 12 months after the conventional therapy. The levels of the PPD mean decreased significantly after treatment with ozone therapy (Graph 4), by 61% at 12 months after the therapy, as compared with control group in which the mean of PPD was decreased by 41% at 12 months after the conventional therapy. The value of CAL is showed in graph 5. In the case group, treated with ozone therapy, the mean of CAL was increased by 15% at 12 months after the therapy, as compared with control group in which the mean of CAL was decreased by 17% at 12 months after the conventional therapy.
21) Clinical findings, 214-217, p8: Values oh Hba1C levels were not significant according to Table 2 in contrast to the claims in “Significant differences among the treatments were evident in CAL and Hba1C level at each time point (P > 0.05). At 3 and 6 months, the case group had slight but significant reductions of glycated hemoglobin level compared with control group, however, there were no significant differences at 12 months (P > 0.05) (Table 2).” Besides mean CAL value of treated patients was already lower (2.025) as compared to that of control patients (2.825) at time 0 making the comparison inadequate.
-It has been corrected as above mentioned.
22) Discussion, line 226, p8 to line 236, p9: Merge into the introduction “The population with type II Diabetes (T2DM) presents a critical challenge in terms of periodontal 226 condition. Diabetic patients are more prone to develop periodontal disease (is estimated to be double 227 the number in the healthy population), and often are affected by severe form of periodontal infection secondary to their underlying systemic alterations and are more likely to be resistant to periodontal treatment than individuals without diabetes [4, 5]. Second, diabetic patients with periodontitis are known to be at elevated systemic inflammatory risk compared to the general population, making periodontal status control an important for this group of patients [6, 7]. The bidirectional relationship between periodontitis and diabetes has been widely discussed in literature [22-26], showing a significant improvement of both diseases after the eradication of periodontal infection. More direct evidence regarding the effects of periodontal disease on glycemic control of diabetic patients comes from intervention studies using periodontal therapy [23].”
-It has been included in the introduction section.
23) Discussion, lines 250-254, p9: Rephrase since it is contradictory. “Our results are according with a previous study conducted by Skurska et al. [31], which demonstrated a significant decrease of periodontal inflammatory indices after the association of periodontal treatment and gaseous ozone, but did not register a greater improvement in these parameters in patients who underwent only scaling and root ”
-It has been corrected as follows:
Our results are according with a previous study conducted by Skurska et al. [31], which demonstrated a significant decrease of periodontal inflammatory indices after the association of periodontal treatment and gaseous ozone compared to the control group.
24) Discussion, lines 254-260, p9: Rephrase since two sentences are contradictory in “Interestingly, Uraz et al [32]. conducted a similar study showing that clinical parameters in patients with periodontitis did not improve with the adjunct of ozone therapy to conventional periodontal treatment. The authors observed that were no significant differences between two treatments groups for any of the microbiological, clinical and biochemical parameters. To our knowledge, no studies have investigated the effect of gaseous ozone as support of periodontal treatment in diabetic patients. For this reason, it is difficult to measure the effectiveness of this therapy and to compare our results.”
-It has been corrected as follows:
In contrast, Uraz et al [32]. conducted a similar study showing that clinical parameters in patients with periodontitis did not improve with the adjunct of ozone therapy to conventional periodontal treatment. The authors observed that were no significant differences between two treatments groups for any of the microbiological, clinical and biochemical parameters.”
25) Conclusion, lines 262-266: The findings were insufficiently convincible to support “In conclusion, gaseous ozone therapy is a noninvasive application, safe and easy to perform, and may be a supporting treatment for periodontitis, having a beneficial effect on the periodontium. Gaseous ozone application following SRP may contribute to improvements in the glycated hemoglobin level, but further longitudinal studies are needed to confirm the effectiveness of ozone application.”
-It has been corrected as follows:
Although our findings suggested that gaseous ozone therapy is a noninvasive application, safe and easy to perform, and may be a supporting treatment for periodontitis in patients with diabetes, having a beneficial effect on the glycated hemoglobin, it is unclear whether the results are generalizable to the larger population. Although there were not statistically differences between specific patient groups, there was a trend toward more effectiveness with gaseous ozone application following SRP. Therefore, currently available evidence suggests that gaseous ozone therapy is effective in reduction of periodontal inflammatory status, but there are insufficient data to reach evidence-based conclusion about the effectiveness on reduction of glycated hemoglobin level in diabetic patients. In conclusion, the results of this study are suggestive, but one cannot arrive a strong conclusion from the results of only one trial. Further longitudinal studies are needed to confirm the effectiveness of ozone application on glycated hemoglobin in diabetic patients.
Reviewer 2 Report
The subject of the article is interesting, it addresses two diseases that are very common in the world population and that has been extensively studied in the sense of finding adjuvant therapies that can improve the performance of conventional treatments.
However, I highlight two problems that I consider important:
1- the cited literature, despite having some classic studies, is not current, most of the articles are from the past decade. An update would be necessary.
2- the authors cite only "Periodontal Disease" showing no criteria for diagnosis and neither the more general classification in gingivitis or periodontitis, or even the new classification proposed in the World Workshop on the Classification of Periodontal and Peri-Implant Diseases and Conditions. J Clin Periodontol. 2018; 45 (February): S219–29.
Author Response
Reviewer #2
The subject of the article is interesting, it addresses two diseases that are very common in the world population and that has been extensively studied in the sense of finding adjuvant therapies that can improve the performance of conventional treatments.
However, I highlight two problems that I consider important:
1- the cited literature, despite having some classic studies, is not current, most of the articles are from the past decade. An update would be necessary.
-The references have been updated. Relating to the references about the ozone therapy, there are a few studies. We have indicated the only significative reports.
2- the authors cite only "Periodontal Disease" showing no criteria for diagnosis and neither the more general classification in gingivitis or periodontitis, or even the new classification proposed in the World Workshop on the Classification of Periodontal and Peri-Implant Diseases and Conditions. J Clin Periodontol. 2018; 45 (February): S219–29.
-It has been added as follows: Based on the new classification of periodontal conditions, a wide range of periodontal diseases can be identified, characterized based on a multidimensional staging and grading system, indicating severity and extent of diseases, which include four categories that also involves systemic diseases and conditions that affect the periodontal supporting tissues, such as diabetes.
Reviewer 3 Report
I am grateful for the possibility to revise this research study.
Ozone Gas Therapy in Conjunction with Periodontal Treatment Improve the Glycated Hemoglobin Level in diabetic people is a trend topic in the current research literature and may be a main focus of interest for readers.
This is a well-written manuscript with an important clinical message, and should be of great interest to the readers of International Journal of Environmental Research and Public Health. Diabetic disorders are a prevalent condition so is very important in order to help readers about a better knowledge of this syndrome.
On the other hand ozone therapy is a useful medical procedure and management of this technique could reduce the complication rates.
Authors should change the tittle, I suggest a shorter tittle to reflect the main of the research, from my point of view to include a question inside the tittle is not a orthodox way to reflect the novelty and the results
Results of the abstract need to reflect the findings with respect to both groups and the lack of significant differences of balance, and also you need reflect the main results obtained more clearly
Introduction section is deep enough with and adequate focus that may help readers to improve knowledge about the topic. However authors should improve the stay of art, for example including references to neuropathic risk and another diabetic complications as the case of diabetic foot releated to HbA1c levels. I suggest to include this references include in the attached to complete this requirement
In line 92
-Chicharro-Luna, E.; Pomares-Gómez, F.J.; Ortega-Ávila, A.B.; Marchena-Rodríguez, A.; Blanquer-Gregori, J.F.J.; Navarro-Flores, E. Predictive model to identify the risk of losing protective sensibility of the foot in patients with diabetes mellitus. Int. Wound J. 2020, 17, 220–227.
Or in the case or complications related to diabetic foot
-Navarro-Flores, E.; Morales-Asencio, J.M.; Cervera-Marín, J.A.; Labajos-Manzanares, M.T.; Gijon-Nogueron, G. Development, validation and psychometric analysis of the diabetic foot self-care questionnaire of the University of Malaga, Spain (DFSQ-UMA). J. Tissue Viability 2015, 24, 24–34.
More over on introduction section I request to authors they should describe better this sentence
Line 71
“Typically, periodontal infection exhibits a wide range of clinical severity that often correlates with the level of glycated haemoglobin (HbA1c)”
I disagree with this affirmation, because, the origen of the problem it is different to correlation . In other word, the fact of the correlation is not necessary the cause of the problem
Methods are well-designed with relevant and complete information. Correct sample size calculations,
However, I suggest to authors they should make a good description of the properties of the outcome measurements, for example in line 116 authors should describe better the meaning of the expression, “concomitant disease “ because in my opinion this expression not reflects clearly the exclusion criteria, as well as improve more detailed statistical analyses in results section
Tables, figures and redaction of the results are presented in a correct way providing a good presentation of the main finding of the study. But on table number 2, authors only highlighted significance results, although most of the research results have a high significance level p value over 0.05
In line 107 to 110 in methods section I suggest authors must include a reference to Ethics requirements Helsinki declaration
-Holt GR. Declaration of Helsinki—The World’s Document of Conscience and Responsibility. South Med J. 2014 Jul;107(7):407–407.
Discussion section is well structured with different sections. Authors manage well the discussion leading a good comparison with the showed references.
However, author should discuss the possible influence of aging a health related quality of life in diabetic patients, as a consequences of their complications due to the disease,in their study finding suggest to include this references include in the attached to complete this requirement because are not included
-Navarro-Flores, E.; Pérez-Ros, P.; FM, M.-A.; Julían-Rochina, I.; Cauli, O. Neuro-psychiatric alterations in patients with diabetic foot syndrome. CNS Neurol. Disord. - Drug Targets 2019, 18.
-Navarro-Flores, E.; Cauli, O. Quality of life in individuals with diabetic foot syndrome. Endocrine, Metab. Immune Disord. - Drug Targets 2020, 20.
Moreover authors should describe better some affirmation, For example, I suggest to review the following sentence in line 236 to 238
“Since the beginning of the 1990s, several studies have investigated the effects of periodontal therapy on the fluctuation of glycated hemoglobin in diabetic patients, demonstrating an improvement in glycemic control, as determined by reductions in HbA1c values, after periodontal treatment”
Conclussions are supported by the shown data.
Author Response
Reviewer #3
Authors should change the tittle, I suggest a shorter tittle to reflect the main of the research, from my point of view to include a question inside the tittle is not a orthodox way to reflect the novelty and the results.
Results of the abstract need to reflect the findings with respect to both groups and the lack of significant differences of balance, and also you need reflect the main results obtained more clearly.
-It has been corrected as follows: Results: At 12 months, the periodontal treatment in conjunction with ozone gas therapy did not showed significant differences than standard therapy in decreasing glycated hemoglobin (HbA1C) level, and is evident the lack of significant differences of balance
Introduction section is deep enough with and adequate focus that may help readers to improve knowledge about the topic. However authors should improve the stay of art, for example including references to neuropathic risk and another diabetic complications as the case of diabetic foot releated to HbA1c levels. I suggest to include this references include in the attached to complete this requirement
In line 92
-Chicharro-Luna, E.; Pomares-Gómez, F.J.; Ortega-Ávila, A.B.; Marchena-Rodríguez, A.; Blanquer-Gregori, J.F.J.; Navarro-Flores, E. Predictive model to identify the risk of losing protective sensibility of the foot in patients with diabetes mellitus. Int. Wound J. 2020, 17, 220–227.
Or in the case or complications related to diabetic foot
-Navarro-Flores, E.; Morales-Asencio, J.M.; Cervera-Marín, J.A.; Labajos-Manzanares, M.T.; Gijon-Nogueron, G. Development, validation and psychometric analysis of the diabetic foot self-care questionnaire of the University of Malaga, Spain (DFSQ-UMA). J. Tissue Viability 2015, 24, 24–34.
-It has been added as follows:
It is well established that diabetic condition encourages development of periodontal disease through a process involving theglucose-mediated advanced glycation end product (AGE) accumulation, that stimulates the gradual transformation of the subgingival microflora into a more pathogenic subgingival flora, responsible of progression of periodontal condition. On the other hand, periodontal infection has been proposed to promote an amplification of the magnitude of the advanced glycation end product (AGE)-mediated upregulation of cytokine synthesis and by chronic stimulus from LPS and products of periodontopathic organisms. These metabolic end products are responsible of the degenerative changes usually observed in diabetic subjects.
More over on introduction section I request to authors they should describe better this sentence
Line 71
“Typically, periodontal infection exhibits a wide range of clinical severity that often correlates with the level of glycated haemoglobin (HbA1c)”
I disagree with this affirmation, because, the origen of the problem it is different to correlation . In other word, the fact of the correlation is not necessary the cause of the problem.
-Well. This phrase has been corrected with the term “associated”, as suggested.
Methods are well-designed with relevant and complete information. Correct sample size calculations, However, I suggest to authors they should make a good description of the properties of the outcome measurements, for example in line 116 authors should describe better the meaning of the expression, “concomitant disease “ because in my opinion this expression not reflects clearly the exclusion criteria, as well as improve more detailed statistical analyses in results section.
-They have been corrected in each session.
Tables, figures and redaction of the results are presented in a correct way providing a good presentation of the main finding of the study. But on table number 2, authors only highlighted significance results, although most of the research results have a high significance level p value over 0.05
In line 107 to 110 in methods section I suggest authors must include a reference to Ethics requirements Helsinki declaration
-Holt GR. Declaration of Helsinki—The World’s Document of Conscience and Responsibility. South Med J. 2014 Jul;107(7):407–407
-It has been included.
Discussion section is well structured with different sections. Authors manage well the discussion leading a good comparison with the showed references.
However, author should discuss the possible influence of aging a health related quality of life in diabetic patients, as a consequences of their complications due to the disease,in their study finding suggest to include this references include in the attached to complete this requirement because are not included
-Navarro-Flores, E.; Pérez-Ros, P.; FM, M.-A.; Julían-Rochina, I.; Cauli, O. Neuro-psychiatric alterations in patients with diabetic foot syndrome. CNS Neurol. Disord. - Drug Targets 2019, 18.
-Navarro-Flores, E.; Cauli, O. Quality of life in individuals with diabetic foot syndrome. Endocrine, Metab. Immune Disord. - Drug Targets 2020, 20.
- The references have been included.
Moreover authors should describe better some affirmation, For example, I suggest to review the following sentence in line 236 to 238
“Since the beginning of the 1990s, several studies have investigated the effects of periodontal therapy on the fluctuation of glycated hemoglobin in diabetic patients, demonstrating an improvement in glycemic control, as determined by reductions in HbA1c values, after periodontal treatment”
Round 2
Reviewer 1 Report
The authors addressed adequately my concerns.
Reviewer 2 Report
The requested review is adequate, within the limits of the study and therefore I believe that the article can be published